# Practical RGB-to-XYZ Color Transformation Matrix Estimation under Different Lighting Conditions for Graffiti Documentation

**DOI:** 10.3390/s24061743

**Published:** 2024-03-07

**Authors:** Adolfo Molada-Tebar, Geert J. Verhoeven, David Hernández-López, Diego González-Aguilera

**Affiliations:** 1Department of Cartographic and Land Engineering, Higher Polytechnic School of Avila, University of Salamanca, Hornos Caleros 50, 05003 Avila, Spain; admote@usal.es; 2Department of Prehistoric and Historical Archaeology, University of Vienna, Franz-Klein-Gasse 1, 1190 Vienna, Austria; geert.verhoeven@univie.ac.at; 3Institute for Regional Development (IDR), University of Castilla La Mancha, 02071 Albacete, Spain; david.hernandez@uclm.es

**Keywords:** colorimetry, color science, graffiti art, image processing, software development

## Abstract

Color data are often required for cultural heritage documentation. These data are typically acquired via standard digital cameras since they facilitate a quick and cost-effective way to extract RGB values from photos. However, cameras’ absolute sensor responses are device-dependent and thus not colorimetric. One way to still achieve relatively accurate color data is via camera characterization, a procedure which computes a bespoke RGB-to-XYZ matrix to transform camera-dependent RGB values into the device-independent CIE XYZ color space. This article applies and assesses camera characterization techniques in heritage documentation, particularly graffiti photographed in the academic project INDIGO. To this end, this paper presents COOLPI (COlor Operations Library for Processing Images), a novel Python-based toolbox for colorimetric and spectral work, including white-point-preserving camera characterization from photos captured under diverse, real-world lighting conditions. The results highlight the colorimetric accuracy achievable through COOLPI’s color-processing pipelines, affirming their suitability for heritage documentation.

## 1. Introduction

Accurately documenting color is necessary for correct color identification, dissemination and reproduction, especially in cultural heritage studies [1,2,3]. Nowadays, heritage documentation relies almost exclusively on digital techniques, in most cases through photo acquisition and image processing procedures [4,5,6]. The problem is that the RGB values recorded by the silicon sensor of consumer digital cameras are device-dependent, and none of them satisfy the Luther–Ives condition; i.e., the RGB values registered by the camera sensor are not a linear combination of the color-matching functions of the standard observer defined by the Commission Internationale de l’Éclaraige (CIE) [7]. In addition, the RGB data are dependent on the scene illumination. If an image is captured in its final form (as a JPG or TIFF file), the RGB values also depend on the image processing parameters set in the camera. Therefore, color values provided by any digital camera cannot be considered strictly colorimetric [8,9].

Given that many documentation and preservation techniques for cultural assets are based on digital images, it is necessary to implement procedures that can turn the camera-specific pixel values into reasonably color-accurate output RGB values. One widely used approach used to achieve this is camera characterization, a procedure yielding a bespoke RGB-to-XYZ matrix. This matrix can then transform the device-dependent RGB data recorded by the camera into the camera-independent, human-vision-based color space CIE XYZ [10,11]. Afterwards, these CIE XYZ tristimulus values can be transformed into any possible RGB output color space like sRGB or Adobe RGB (1998) via well-known procedures.

Different methodologies have been developed to estimate this RGB-to-XYZ transformation matrix [7,12,13,14,15,16,17,18,19]. For all of them, it is highly recommended to use RAW RGB values for the computations since RAW data tend to be linear to the image radiance [11,20]. In addition, RAW data are not subject to the pre-processing that in-camera-generated JPG or TIFF images undergo, which, in most cases, is applied automatically with little control over the process [21,22,23]. A linear regression model suffices to compute the matrix since the relationship between the RAW RGB values and CIE XYZ tristimulus values is expected to be linear [13]. After the linear regression adjustment, a 3 × 3 color correction matrix or camera characterization matrix is obtained that characterizes the RGB-to-XYZ transformation.

Using linear regression models for camera characterization is advantageous for multiple reasons: they are easy to interpret even for non-expert users, are computationally cost-effective, and provide relatively accurate results. However, a significant drawback is their dependency on the data used. The transformation matrices are illuminant-dependent since the relative RAW RGB values are highly sensitive to changes in the spectral distribution of the scene illumination. Therefore, applying RGB-to-XYZ matrices should be restricted to scenarios where an object is photographed under the same illumination conditions as those used to acquire data for the camera characterization procedure. If there is a change in illumination, a new transformation matrix must be calculated for that particular illuminant and that 3 × 3 matrix should not be used for a different illuminant. From a practical point of view, this technique is unfeasible, especially in heritage documentation projects where many photographs acquired under different illumination conditions must be handled.

One example of such a case is INDIGO (INventory and DIsseminate Graffiti along the dOnaukanal), a two-year academic project launched in September 2021 through funding from the Heritage Science Austria program of the Austrian Academy of Sciences. Project INDIGO wanted to push the status quo boundaries in inventorying and understanding extensive graffiti-scapes. This was mainly achieved by acquiring photos of new graffiti via weekly photographic tours, leading to a massive volume of digital photos obtained in widely varying illumination conditions [24,25,26].

One way to address the problem is to perform the RGB-to-XYZ transformation under the assumption of homogeneous D65 illumination. Normally, the standard illuminant D65 defined by the CIE should be used in all colorimetric calculations. D65 represents average daylight and has a correlated color temperature of approximately 6500 K [10]. Previous investigations in rock painting scenarios indicated that satisfactory results were obtained using second-order polynomial models for colorimetric camera characterization [11,27,28]. However, in practice, it is not possible to work under controlled lighting conditions, especially not in outdoor photography where there is rarely a single reference white point present. Natural outdoor scenes also feature different states of adaptation and viewing conditions because of the simultaneous appearance of shadowed areas and regions with direct incident sunlight.

The present study aims to obtain an accurate RGB-to-XYZ transformation matrix from photos acquired under different illumination conditions. To achieve this objective, an analysis of the well-established white point preservation (WPP) technique has been carried out. The WPP is a variant of a linear least square regression model in which a constraint is added to preserve the reference white point along with the grays [13]. This constraint makes it possible to obtain RGB-to-XYZ transformation matrices that are more robust to changes in illumination.

The present research was split into the following tasks to achieve this objective:Assessment of the WPP technique under different illumination conditions;Evaluation of the influence of color charts from different manufacturers on this process;Comparison of an ordinary least squares (OLS) model versus a set of linear regression algorithms;Analysis of the color accuracy achieved by applying the RGB-to-XYZ transformation matrix to a real-world scenario with graffiti photos.

As a novel aspect of this study, the analysis and processing of the digital photos have been carried out entirely using the functionalities implemented in the COlor Operations Library for Processing Images or COOLPI package [23,29,30]. COOLPI was created within the INDIGO project. Since one of the fundamental pillars of the INDIGO project was the creation of color-accurate digital orthophotos, the COOLPI package integrates classes, methods, and functions for color image correction.

## 2. Materials and Methods

### 2.1. RGB-to-XYZ Color Transformation Matrix Computation

Different digital camera devices have different opto-electronic conversion functions and spectral sensitivities. Manufacturers typically do not provide this information, and the most accurate and correct way to obtain the spectral sensitivities of a camera is through specific instrumentation (monochromators and spectroradiometers) under strict laboratory conditions [31]. The problem is that these instruments are unavailable to the general public, so an alternative methodology must be applied to achieve color-accurate data from photos.

From a practical point of view, a widely used approach is the RGB-to-XYZ transformation matrix estimation based on a target-based method [32]. This methodology establishes the mathematical relationship between the RGB data of a set of samples recorded by the camera and their CIE XYZ values measured by a colorimeter or preferably a spectrophotometer. To obtain the RGB- to-XYZ transformation matrix coefficients, two different sets of data are required: the CIE XYZ values (computed from the spectral data) of the patches of a color reference chart (like the famous ColorChecker) used as colorimetric reference; and the RGB data registered by the camera extracted from a digital image of the same color patches. Also, the illuminant under which both measurements were taken must be considered.

### 2.2. White Point Preservation Constraint

The RGB values acquired by a digital camera are highly sensitive to the incident spectral radiance, so they will vary according to the illuminant under which the photograph was taken. As a direct consequence, the RGB-to-XYZ transformation matrix coefficients obtained vary depending on the data used for training the model [7].

One way to address the variation in the RGB-to-XYZ matrix is via the WPP constraint [12,13], which normalizes the RGB and XYZ data into the range [0, 1] using the reflectance closest to a perfect white diffuser. Effectively, all RAW RGB and CIE XYZ values are divided by the RGB and XYZ coordinates corresponding to a white patch on a color reference target. In this way, any pixel with the maximum possible RGB values (1,1,1) in the RAW image will map onto CIE XYZ (1,1,1) values, which implies that the whites (and the grays) are preserved regardless of the illuminant [13]. In this way, obtaining more robust and stable transformation matrices in the case of illumination changes is possible.

### 2.3. Spectral Data

Color reference targets are widely used tools for digital imaging applications, particularly for color calibration. However, different reference targets, provided by different manufacturers, exist. The most widespread targets are probably those provided by X-Rite (now Calibrite) and Datacolor [33,34,35]. For that reason, the color reference targets selected for this study as a colorimetric reference are as follows:Calibrite ColorChecker Digital SG (CCDSG, 96 patches or color samples when the achromatic patches from the edges are removed);Calibrite ColorChecker Classic (CCC, 24 patches);X-rite ColorChecker Passport Photo 2 (XRCCPP, 24 patches);Datacolor Spyder Checkr (SCK100, 48 patches).

The reflectance data of the color patches were measured using a Konica Minolta CM600d spectrophotometer (Konica Minolta, Spain, Valencia) with the SpectraMagic NX software v.3.31. The CIE XYZ tristimulus values for the standard D65 illuminant were computed in COOLPI directly from the reflectance data acquired by the spectrophotometer using the CIE formulation by integrating the product of the spectral reflectance, the spectral power distribution (SPD) of the D65 illuminant, and the color-matching-functions for the 2° observer (a mathematical representation of the average color vision of humans) [10,23].

### 2.4. Illuminants

The digital photos were acquired in two scenarios: laboratory or controlled illumination conditions and in situ illuminants (corresponding with indoor and outdoor illuminants). The shots under controlled lighting took place in a Just Normlicht Color Viewing Light S PROFESSIONAL (JNCVLS) color cabin. The JNCVLS integrates illuminants approximately corresponding to the CIE standard D65, A, F11 (or TL84), and D50 [36]. Theoretical SPD data for standard illuminants are provided by the CIE [10]. However, it is highly recommended that actual measurements of the cabin illuminant be taken at the time of the photo. Thus, the SPD for all color cabin illuminants were measured using a Sekonic C-7000 SPECTROMASTER spectrometer (SEKONIC, Austria, Viena) and labeled using the prefix JN-, indicating that it is a color cabin illuminant, followed by the nomenclature of the standard illuminant, i.e., JN-D65, JN-A, JN-F, and JN-D50, respectively. 

For photographs taken under uncontrolled lighting conditions, the SPDs of the illuminants were also measured at the time of image capture and labeled with the prefixes IN- and OUT- for indoor and outdoor, respectively, followed by the number identifying the measurement made with the spectrometer. In total, three different in situ illuminants were considered: IN-29, corresponding to artificial indoor lighting; IN-30, for indoor ambient natural light; and OUT-38, for natural daylight or direct sunlight outdoor illumination. Figure 1 shows the SPD of the illuminants obtained with the spectrometer. In addition, Table 1 mentions the correlated color temperature (CCT) in K and the white point *X*_n_, *Y*_n_, and *Z*_n_ tristimulus values (CIE 1931 2° observer) for each illuminant, as these values are useful in defining an illuminant in addition to its SPD.

### 2.5. Digital Photos

Two different digital cameras were used for this study: a Nikon D5600 (Nikon Corporation, Spain, Valencia) and a Nikon Z 7II (Nikon Corporation, Austria, Vienna), the latter being one of the official INDIGO cameras [24]. The exposure time, ISO, and aperture were controlled for all the photos to ensure proper exposure. As a sample, Figure 2 displays the images corresponding to the XRCCPP reference target for the Nikon D5600 and Figure 3 for the Nikon Z 7II camera. Since both cameras allow saving photos as NEF files (i.e., Nikon Electronic Format, Nikon’s RAW image file format), these were used in this study. The RAW data were black-level corrected, demosaiced with COOLPI’s default adaptive homogeneity-directed demosaicing algorithm (i.e., to ensure every pixel had R, G, and B values), and scaled in the range [0, 1]; no white balancing was applied. The resulting data are henceforth referred to as the original data (unconstrained). These photos supplied the camera- and illumination-specific RAW RGB data used to calculate the coefficients of the RGB-to-XYZ transformation matrices, a procedure detailed in Section 2.7.

For the Nikon D5600, a total of 28 images were available: 16 were acquired under controlled (or laboratory) lighting conditions: 1 for each combination of cabin illuminant (JN-D65, JN-A, JN-F, and JN-D65; Section 2.4) and color reference target (CCDSG, CCC, XRCCPP, and SCK100; Section 2.3); the remaining 12 were taken under in situ illuminants: 1 for each combination of in situ illuminant (IN-29, IN-30, and OUT-38; Section 2.4) and reference target (CCDSG, CCC, XRCCPP, and SCK100; Section 2.3). The D5600 photos were used to compute the transformation matrix coefficients using the WPP constraint from a theoretical perspective and compared with the results achieved using the original data (see Section 2.6).

However, since it is generally impossible to work under controlled lighting conditions, there was an interest in evaluating the accuracy achieved in determining the transformation matrix using only photos taken under in situ illumination, and its application to graffiti photos as part of the INDIGO project. To that end, 12 Nikon Z 7II photos acquired in situ under natural illumination were used to estimate the RGB-to-XYZ matrix: 1 for each combination of in situ illuminant (IN-29, IN-30, and OUT-38; Section 2.4) and color reference target (CCDSG, CCC, XRCCPP, and SCK100; Section 2.3). After obtaining the matrix coefficients, they were applied to 45 photos of different graffiti located along the Donaukanal in Vienna, Austria.

### 2.6. Three Model Datasets

Section 2.5 clarified that this study used 28 Nikon D5600 photos and 12 Nikon Z 7II photos. From this photo collection, three model-specific datasets were generated. These datasets are labelled with the prefix ND- or NZ- indicating the camera used (i.e., the Nikon D5600 and Z 7II, respectively) followed by the suffix ODS (for the original, unconstrained data) or WPPDS (for the data considering the WPP constraint). 

In addition to the number of images used for each dataset, Table 2 details the color samples (i.e., the patches of the reference chart) used, considering the different illuminants and color reference targets described in Section 2.5. The ‘Laboratory Photos’ column refers to images acquired under controlled lighting conditions (illuminants JN-D65, JN-A, JN-F, and JN-D65; Section 2.4) for each of the color reference targets considered (CCDSG, CCC, XRCCPP, and SCK100; Section 2.3), while the ‘In situ Photos’ column refers to images acquired under indoor and outdoor illuminants (IN-29, IN-30, and OUT-38; Section 2.4), for the same color reference targets. These datasets were used in the linear regression fitting to obtain the RGB-to-XYZ transformation matrix coefficients for each camera. 

Using two datasets for the Nikon D5600 has a threefold purpose. First, it allows the evaluation of the improvement achieved by including the WPP constraint in the OLS adjustment (ND-ODS vs. ND-WPPDS). Second, different regression models can be compared. Finally, and this is one of the most important aspects of this study, it was possible to analyze whether the coefficients of the transformation matrix obtained after adjustment (from the ND-WPPDS) in outdoor lighting are comparable to those obtained in laboratory conditions.

For the ODS dataset, the 2° XYZ and RAW RGB values were scaled to the range (0, 1) as follows:(1)$$RGB=\left\{\frac{R}{tonal\;range},\frac{G}{tonal\;range},\frac{B}{tonal\;range}\right\}$$
(2)$$tonal\;range={(2}^{bit\;depth}-1)$$
(3)$$XYZ=\left\{\frac{X}{100},\frac{Y}{100},\frac{Z}{100}\right\}$$
in which tonal range refers to the image’s potential different tones, a number defined by the bit depth of the image (e.g., a 12-bit RAW image maximally contains 2^12^—1 or 4095 tonal levels).

For the WPPDS dataset, the RAW RGB and 2° XYZ values were normalized using the patch with the maximum XYZ and RAW RGB value (white point preservation constraint, Section 2.2), which corresponds with the white patch of the color reference target as follows:(4)$$RGB=\left\{\frac{R}{R_{wpp}},\frac{G}{G_{wpp}},\frac{B}{B_{wpp}}\right\}$$
(5)$$XYZ=\left\{\frac{X}{X_{wpp}},\frac{Y}{Y_{wpp}},\frac{Z}{Z_{wpp}}\right\}$$
where R_wpp_, G_wpp_, B_wpp_, X_wpp_, Y_wpp_, and Z_wpp_ refer to the white patch RAW RGB and 2° CIE XYZ values, respectively.

### 2.7. Linear Regression Model Comparison

The aim was to find the mapping function between the RAW RGB color values registered by the digital camera (device-dependent) and their corresponding CIE XYZ tristimulus values (device-independent). Different regression models can be applied to compute this transformation [14,27,37,38,39,40,41]. The number of parameters of the transformation will depend on the regression model used. However, since RAW RGB data tend to be linear, using linear regression models is accurate enough, so a 3 × 3 RGB-to-XYZ transformation matrix will be obtained. 

The normalized RAW RGB and CIE XYZ triplets (see Section 2.6) can be expressed as a positional vector as follows:(6)$${RGB}_{RAW}=\{R,G,B\}$$
(7)XYZ={X, Y, Z}

Denoting M as the RAW RGB to CIE XYZ 3 × 3 transformation matrix, we can express the mathematical model as:(8)$${\left[XYZ\right]}_{3\times n}=M_{3\times 3}\cdot {\left[RGB\right]}_{3\times n}=\left|\begin{array}{ccc} m_{00} & m_{01} & m_{02}\\ m_{10} & m_{11} & m_{12}\\ m_{20} & m_{21} & m_{22} \end{array}\right|\cdot {\left[RGB\right]}_{3\times n}$$
where the subindex n denotes the number of samples used for training the model. 

The model can be adjusted using an ordinary least squares regression (OLS). The least square regression determines the model’s coefficients so that the sum of the squared residuals is at a minimum. The entire OLS process can be divided into the following stages:Loading the model data (i.e., the RAW RGB and CIE *XYZ* triplets of the color patches).Splitting these patches into two randomly selected datasets: training (80%) and testing (20%).OLS model regression adjustment using the training data to obtain the coefficients for the transformation matrix.Model assessment from test data. The RMSE and the CIE *XYZ* residuals for the predicted values were computed. Also, color difference metrics were calculated to evaluate the color accuracy obtainable via the computed transformation matrix (Section 2.8).

In addition, a comparison between different linear regression models was carried out to examine whether improving the regression model in terms of results and computational effort is possible. The OLS model was compared with the following robust and well-established models: multi-task Lasso CV (MTLCV); Bayesian ridge regression (BR); the Huber regressor (HR), and the Theil–Sen regressor (TSR). The model comparison was performed using the Scikit-learn Python package, so details about these different regression models can be found in the Scikit-learn API documentation [42].

### 2.8. Model Assessment

The model assessment metrics were computed using the test data. Since RMSE and residuals are in CIE XYZ units and do not necessarily reflect perceived visual errors, and since the CIELAB color space is approximately perceptually uniform, color difference metrics have been included to assess the colorimetric accuracy achieved after the adjustment [11,13,43]. Thus, for calculating color differences, the 2° CIE XYZ predicted values must be transformed into the CIELAB color space (under illuminant D65) using the CIE formulation [10]. Conversion between color spaces is well-documented in the literature. We recommend following the method described in reference [11] for a detailed analysis. The CIELAB space characterizes colors according to three parameters: *L** for luminance, *a** for the green–red chromaticity axis, and *b** for the blue–yellow chromaticity axis. Two different color difference metrics were used: the legacy but often reported ${∆E}_{ab}^{*}$ and the improved ${∆E}_{00}$ or CIEDE2000 [10], which is the current industry standard. 

Given a pair of color stimuli defined in CIELAB space, the color difference ${∆E}_{ab}^{*}$ between them can be computed as follows [10]:(9)$${∆E}_{ab}^{*}=\sqrt{{\left(∆L^{*}\right)}^{2}+{\left(∆a^{*}\right)}^{2}+{\left(∆b^{*}\right)}^{2}}$$
where $∆L^{*}$, $∆a^{*}$, and $∆b^{*}$ are the differences between the *L**, *a**, and *b** coordinates of the two-color stimuli. A difference value of 0 indicates that the colors are identical, and higher values indicate increasing perceptual differences between the colors. 

However, it is worth noting that ${∆E}_{ab}^{*}$ has some limitations and may not perfectly match human perception in all cases. Therefore, more advanced color difference formulas, such as ${∆E}_{00}$ have been developed to address some of these limitations. ${∆E}_{00}$ is a color difference equation recommended by the CIE in 2001 that improves the computation of color differences for industrial applications [44,45,46]. The ${∆E}_{00}$ equation corrects the non-uniformity of the CIELAB color space, adding lightness, chroma, and hue weighting functions and a scaling factor for CIELAB *a** axis [44,47,48]. 

The ${∆E}_{00}$ color difference equation must be applied as follows [10,47,48]:(10)$${∆E}_{00}=\sqrt{{\left(\frac{∆L\prime }{k_{L}{\cdot S}_{L}}\right)}^{2}+{\left(\frac{∆C\prime }{k_{C}{\cdot S}_{C}}\right)}^{2}+{\left(\frac{∆H\prime }{k_{H}{\cdot S}_{H}}\right)}^{2}+R_{T}\cdot \left(\frac{∆C^{\prime }}{k_{C}{\cdot S}_{C}}\right)\cdot {\left(\frac{∆H\prime }{k_{H}{\cdot S}_{H}}\right)}^{2}}$$
where the S_L_, S_C_, S_H_, and R_T_ are the weighting functions, and k_L_, k_C_, and k_H_ are the parametric factors or correction terms for variations in experimental conditions (under reference conditions, they are set to k_L_ = k_C_ = k_H_ = 1 [10]). 

Given the intricacies inherent in the computation of ${∆E}_{00}$ color differences, this article provides solely the pertinent equation facilitating its determination (Equation (10)). For an exhaustive study of each of the parameters involved in the calculation of ${∆E}_{00}$, the reader is referred to the specialized colorimetric references, which give a more detailed account of their calculation and the relevant mathematical observations [47]. It is imperative to note that these references have been meticulously considered during the integration of the ${∆E}_{00}$ equation into the COOLPI framework. 

In order to evaluate the quality of the model fit, both the RMSE and the mean of the residuals in CIE XYZ units were analyzed, as well as the values of the color differences ${∆E}_{ab}^{*}$ and ${∆E}_{00}$ to enable the adjustment’s evaluation from a colorimetric point of view. Different criteria can be considered in defining a maximum acceptable color difference, and these thresholds are often goal-specific. For example, Vrhel and Trussell set the ${∆E}_{ab}^{*}$ threshold at ≤3 for color correction purposes [49], while Song and Luo [50] stablished ${∆E}_{ab}^{*}$ ≤ 4.5 as the acceptability threshold when viewing images on a monitor. The digitization advice provided by the Metamorfoze Preservation Imaging Guidelines [51] mentions an average ${∆E}_{ab}^{*}$ color difference of ≤2.83 for the neutral patches and ≤4 for all patches of a CCDSG target. Using the latter guidelines—and in agreement with our experience in rock art documentation [11]—this paper uses an average ${∆E}_{ab}^{*}$ threshold of ≤4. Although values from different color difference metrics should not be compared (they vary across the CIELAB colour space), and even though ${∆E}_{00}$ thresholds are usually tighter than their ${∆E}_{ab}^{*}$ counterparts (see [52] or a comparison of both metrics in dentistry), this paper adopts an average ${∆E}_{00}$ threshold of ≤4. The fact that the ${∆E}_{00}$ threshold is identical to the ${∆E}_{ab}^{*}$ threshold is not only for ease of use, but also because it is in line with the ${∆E}_{00}$ − ${∆E}_{ab}^{*}$ regression equation proposed by Lee and Powers for colour differences below 5 [53].

### 2.9. Methodological Application in a Real Scenario

Typically, cultural heritage photo acquisition occurs under non-homogeneous and uncontrolled illumination. This is certainly the case when photographing cultural assets, like graffiti, placed outside. From the dataset obtained using the images from the Nikon Z 7II camera, the transformation matrix coefficients were calculated using the WPP constraint (Section 4). The color correction matrix was then applied to photos of graffiti located along Vienna’s Donaukanal. To comprehend the outdoor lighting conditions, every graffito photo was complemented by a measurement of the solar spectral illumination (i.e., the illumination’s SPD) using a Sekonic C-7000 SPECTROMASTER (Section 2.4). 

All the steps of the processing pipeline—which was entirely executed within COOLPI (Section 2.10)—can be summarized as follows (the workflow diagram is shown in Figure 4):Load and minimally process (see Section 2.5) the RAW graffito sample photo, which includes an XRCCPP as a colorimetric reference;Obtain the measured SPD of the illuminant corresponding to the graffito photo;Compute the SPD’s CIE XYZ values and multiply them by the inverse of the transformation matrix to yield the white balance multipliers;Apply the multipliers to compute the white-balanced graffito photo;Apply the RGB-to-XYZ color transformation to the white-balanced graffito photo;Transform the CIE XYZ to sRGB to obtain a 16-bit color-corrected TIFF image;Extract the RAW RGB data of the patches of the XRCCPP (for color accuracy evaluation);Assess the color accuracy achieved after the process (computing the metrics described in Section 2.8).

### 2.10. COOLPI

Since one of INDIGO’s main goals was to obtain color-accurate orthophotographs, the COOLPI toolbox was developed within the project. COOLPI is an open-source and completely free Python package, including classes, methods, and functions for colorimetric and spectral data treatment [29,30]. The package has been developed and tested rigorously following the colorimetric standards published by the CIE [8]. In addition, the COOLPI toolbox includes functionalities for RAW image conversion, facilitating control over every RAW development step (something that commercial image processing software tends to hide from the user). COOLPI also comes with a graphical user interface (GUI) that integrates the main functionalities of the COOLPI package. The GUI was made to be intuitive and user-friendly, especially for non-programmers (Figure 5).

## 3. Results and Discussion

### 3.1. White Point Preservation Assessment

As a starting point for this study, we analyzed the behavior of the WPP constraint for computing the RGB-to-XYZ transformation matrix. This analysis was carried out for the two scenarios, i.e., laboratory and in situ illumination. Therefore, the ND-ODS and ND-WPPDS datasets of the Nikon D5600 images (Table 2) were used to compute and perform the model comparison.

Although it is assumed that a camera’s RGB values are linearly related to the incident radiance, it is best to check this assumption. One way of doing this is by plotting the RGB values of the gray patches on the photographed color reference target against their measured CIE XYZ tristimulus values. Figure 6 depicts the outcome of this process for the Nikon D5600 using the XRPPCC and the CCDSG color reference targets. Both graphs show that no non-linearity correction is needed for the RAW RGB data [54].

To determine the color reference target most suitable for obtaining the RGB-to-XYZ matrix, a pre-analysis applied an OLS regression model for the ND-WPPDS dataset to all reference targets under the JN-D65 illuminant (see Table 3). The training and test columns refer to the color patches used for model training and evaluation, respectively. These patches were randomly selected using the train_test_split function of the scikit-learn Python library. As mentioned in Section 2.8, the RMSE and the mean of the residuals (in CIE XYZ units) were analyzed, as well as the ${∆E}_{ab}^{*}$ and ${∆E}_{00}$ color differences. 

${∆E}_{00}$ (Equation (10)) values less than five were obtained for all targets, with higher values for color reference charts featuring more patches. The lowest ${∆E}_{00}$ value (1.642) was obtained with the XRCCPP target, although the model was trained with the fewest data points. In addition, that reference chart also resulted in the highest predictive capability (R^2^ value of 0.968) (Table 3). As expected, the OLS regression model proved accurate enough for obtaining the transformation matrix coefficients, given the linearity of the RAW RGB data.

After selecting the best-performing color reference chart, the comparative analysis between the ND-ODS and ND-WPPDS datasets was carried out using only the XRCCPP data under each illuminant as training/test data. 

Table 4 shows the illuminant-specific model assessment metrics obtained after the adjustment using the ND-ODS and NP-WPPDS datasets. In addition, the results are grouped into “Laboratory” (JN-D65, JN-A, JN-F, and JN-D50), “In situ” (IN-29, IN-30, and OUT-38), and “All” illuminants. The computed RGB-to-XYZ transformation matrix coefficients are shown in Table 5 for the ND-ODS and ND-WPPDS datasets, again for each illuminant separately and grouped into “Laboratory”, “In situ” or “All”.

These analyses show that similar results are obtained using the ND-ODS or ND-WPPDS datasets. Both datasets perform with sufficient accuracy under laboratory conditions and in situ lighting. Values lower than two were obtained for the ${∆E}_{00}$ color difference (Table 4). As expected, significant distinctness is found when comparing the results between the two datasets when grouping illuminants. For the ND-ODS dataset, ${∆E}_{00}$ values greater than four were obtained, while they were always less than four for the ND-WPPDS data. ${∆E}_{00}$ values below four (i.e., the threshold value established for color differences; Section 2.8) guarantee that it is impossible to perceive the difference between the predicted and observed data for the color patches used for model testing, i.e., between the patches in the real world and the color-corrected digital image.

Although from a statistical point of view it can be stated that both datasets give satisfactory results, we are mainly interested in comparing the transformation matrix coefficients obtained with or without the WPP constraint. Large coefficient differences can be observed for the ND-ODS dataset, where it is clear that they depend on the illumination used to obtain them. The opposite situation occurs with the ND-WPPDS dataset. In this case, the coefficients obtained are more homogeneous across illumination changes thanks to the WPP constraint (Table 5).

The ND-WPPDS data yield higher values for the RMSE and residual CIE XYZ values than the ND-ODS data, because the former feature the constraint that the maximum value for CIE XYZ is (1,1,1). Although the resulting matrix preserves the white and grays, it will not work well for pixels whose tristimulus values exceed (1,1,1). For both datasets, higher RMSE values were obtained when the regression used the data for all illuminants. However, it should be noted that higher values were also obtained for R^2^, which implies a higher predictive capability for the model. The highest predictive capabilities were found in the WPP-constrained data. Overall, it is thus clear that the WPP constraint provides better and more stable computation of the coefficients in the RGB-to-XYZ transformation matrix.

The next step was to evaluate the behavior of the WPP constraint on large datasets. Thus, an additional OLS adjustment was performed using the data for all of the color reference charts photographed under each illuminant, and one for all in situ, laboratory, and laboratory + in situ photographs. The model assessment metrics computed after the adjustment can be found in Table 6; Table 7 lists the coefficients obtained for the color transformation matrix.

In view of the results obtained, it can be seen that the combination of color charts performs very well under in situ illuminants. In fact, better results were obtained compared to the model trained using only the JNCVLS color cabin photos where the highest RMSE (0.058) and ${∆E}_{00}$ value (5.067) were achieved (Table 6). This is due to the limitations of the color cabin, which cannot be entirely closed. Also, a larger dataset was used for model training and validation compared to the adjustment using the images under in situ illumination (Table 6, row ‘All’). However, the fit can be considered sufficiently accurate. Regardless of the illuminant used, the coefficients of the transformation matrix exhibit minimal variation (Table 7).

For practical purposes, the results obtained for the in situ illuminants are equivalent to those obtained using the entire WPPDS dataset, in terms of the model predictive capability and color difference metrics, since ${∆E}_{00}$ color differences less than four are sufficiently accurate.

As a supplementary aid to the statistical model metrics presented in Table 6, Figure 7 shows the predicted versus the observed CIE XYZ values, and the residuals obtained after the adjustment using the full ND-WPPDS dataset, while Figure 8 shows the ${∆E}_{00}$ color difference obtained. Observing the graphs, we can affirm that the OLS regression model is adequate to carry out the regression, obtaining satisfactory and accurate results from a colorimetric point of view. Thus, this implies that the RGB-to-XYZ transformation matrix coefficients can be estimated with sufficient accuracy by the model, for laboratory and/or in situ illumination.

### 3.2. Linear Regression Model Comparison

After analyzing the behavior of the OLS regression model, the question is whether it is possible to improve the results by employing more robust and complex linear models, such as those provided by specific mathematical computing libraries like Scikit-learn (which is very versatile and easy to implement and understand).

As stated in Section 2.7, a comparison between the OLS, MTLCV, BR, HB, and TSR regression models was carried out. This model comparison was performed using the full ND-WPPDS dataset from the Nikon D5600 images (Table 2).The results of the comparative analysis are shown in Table 8, and the coefficients for the transformation matrix are shown in Table 9.

The best result was obtained using the HR model, since this model is robust to outliers. However, as can be seen, there were no major improvements compared to OLS. On the other hand, the MTLCV method (which includes a leave-one-out cross-validation) produced identical RMSE and R^2^ compared to an OLS regression, but with slightly higher color differences, so an OLS adjustment including cross-validation is not necessary. The predictive capability of the models (R^2^ around 0.97), as well as the average color differences obtained after fitting (${∆E}_{00}$ less than four), are all equivalent (Table 8). This can also be noted by comparing the coefficients obtained for the transformation matrix (Table 9). The greatest difference is observed in the coefficients obtained via the TSR model, but these differences are not significant from a practical perspective.

From a computational standpoint, there are no major distinctions either. It should be noted that the BR, HR, and TSR models must be calculated independently to obtain the coefficients of the transformation matrix (i.e., three different regression models are needed: RAW RGB to X, RAW RGB to Y, and RAW RGB to Z). However, their implementation in Python is not complicated; in terms of computational cost, there is no difference (except for models trained on large datasets, in which case it would have to be considered by the user).

Although models more robust to outliers (such as HR) can be used, this analysis showed that using the OLS model is appropriate: it is straightforward to interpret, easy to implement, and computationally effective, while providing results comparable to those of more complex models. Thus, in view of the comparison performed, other linear regression models did not result in better results.

## 4. Practical Application in a Graffiti Scenario

So far, this paper has shown that one can obtain a reliable RGB-to-XYZ transformation matrix via an OLS regression model with WPP constraint using Nikon D5600 images taken under different lighting conditions: that is, laboratory settings and/or in situ illuminants (e.g., indoor, outdoor, or both). However, how transferrable is this approach to another camera like Nikon’s Z 7II, and what do the results look like when that computed characterization matrix is applied to a collection of graffiti photographs taken during typical heritage documentation activities?

First, the Nikon Z7 II transform matrix was computed via the OLS regression model using only photos taken under in situ lighting conditions (i.e., the NZ-WPPDS dataset; Table 2), since this is common in cultural heritage documentation projects. The statistical estimators and color differences obtained after the adjustment are given in Table 10 for each illuminant and grouped. Table 11 lists the coefficients of the RGB-to-XYZ transformation matrix.

The best results were obtained for illuminant IN-30, corresponding to indoor ambient light conditions (Section 2.4). Noteworthy are the model’s predictive capability (R^2^ = 0.995) and the average ${∆E}_{00}$ color difference of less than two (Table 10). Even when considering the entire NZ-WPPDS dataset (grouping all illuminants), the predictive capability remained high (R^2^ = 0.985) and ${∆E}_{00}$ stayed below three. The coefficients for each of the four matrices were also similar, as shown in Table 11. Therefore, this methodology for estimating a transformation matrix can be considered appropriate for characterizing different digital photo cameras in various illumination conditions.

Upon obtaining the RGB-to-XYZ transformation matrix for the Nikon Z 7II camera (i.e., the in situ matrix, Table 11), the color transformation was applied to 45 photos of graffiti located along Vienna’s Donaukanal. These photos were acquired by a staff member of project INDIGO during one of the weekly photographic tours. INDIGO’s photographic tours aimed to document new graffiti that had appeared since the last tour. Since almost 13 km of urban surfaces had to be checked during each tour, data were collected relatively quickly and during all weather conditions. This is an important fact to consider because it means the data resulted from a practical, real-world scenario and were not just collected to optimize the colorimetric results. During a photographic tour, each new graffito was documented as follows [26]. First, an XRCCPP target was photographed in front of the graffito, so the target received the same illumination as the graffito. Then, the graffito’s spectral illumination was measured with a Sekonic C-7000 SPECTROMASTER spectrometer. Finally, the graffito was photographed from all possible sides, enabling the creation of a textured 3D surface model and an orthophotograph. This test utilizes the XRCCPP color chart and the illumination’s SPD of 45 new graffiti documented on 22 July 2022.

After following the methodological steps described in Section 2.9, average ${∆E}_{00}$ color difference values for the 24 patches of the XRCCPP chart could be computed. These are given in Table 12, and Figure 9 shows the histogram for the color difference values obtained. Table 13 classifies the absolute and relative number of images according to these average color differences. In addition, Figure 10 depicts a selection of eight original versus color-corrected photos for which the color difference was less than four.

With ${∆E}_{00}$ values of less than four obtained for only half of the photos, these results are less positive than those of the more controlled tests. This is likely due to four factors. First, the data were acquired outdoors in an urban setting. Although it was sunny during photo acquisition, a graffito’s illumination (and thus also that of the reference target) cannot always be homogonous, for example, when located under a bridge (see Figure 11 on the left) or partly in the shade. Such non-homogenous illumination of the target negatively affects achievable color accuracy. Second, the ColorChecker target is held at arm’s length, making it not unthinkable that the photographer partly influenced the target’s illuminations (for example, by partly shielding it or because his clothes reflect on the target). Third, the SPD has been used to compute the white balance multipliers (via the camera transformation matrix; see Section 2.9). The correctness of this approach assumes a suitable camera transformation matrix and identical illumination conditions for the target and the spectrometer measurement. When the latter is compromised, mediocre results will result. Fourth, the target’s patches are not Lambertian reflectors, so a target that is held not perfectly perpendicular to the camera’s optical axis might negatively influence the results (see Figure 11 on the right).

Although data collected during several hours of photographing graffiti in warm conditions is not expected to yield the same colorimetric accuracy as data carefully gathered for academic purposes, all four issues warrant further research so that the entire graffiti documentation workflow—or any heritage workflow for that matter—can be further optimized. Despite these limitations, the current method already allows for computing color-accurate graffiti photographs in half of all cases with a method in which all parameters are controlled and do not result from a black-box procedure.

## 5. Conclusions

Although it is advisable to carry out photographic measurements under laboratory conditions, this requirement can rarely be met, especially in outdoor photography, as is often the case in cultural heritage documentation. This study, which used a white point preservation (WPP) constraint to characterize digital cameras and compute an RGB-to-XYZ transformation matrix, revealed that the color differences achievable via in situ illuminants were largely similar to those obtained under laboratory settings. Furthermore, the results underscored the efficacy of the WPP constraint in camera characterization and revealed that the employed ordinary least squares regression fitting yields results that are comparably accurate to those of more robust but computationally more demanding fitters.

Its application to graffiti photos highlighted that the proposed workflow is suitable for generating color-accurate results in real-world heritage documentation scenarios using different lighting conditions. Although a few aspects of the graffiti documentation pipeline need improvements, these corrections likely have less to do with the developed method and more with the data acquisition. Notably, the workflow has been seamlessly integrated into the COOLPI package, thus supporting the straightforward retrieval of color-accurate data from digital images. The authors are confident that the methodology and its application through COOLPI are suitable for all fields where accurate color information is required.

## Figures and Tables

**Figure 1 sensors-24-01743-f001:**
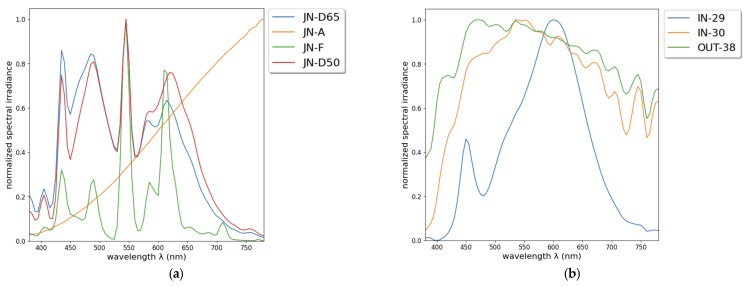
SPD illuminants: (**a**) Laboratory settings. (**b**) In situ.

**Figure 2 sensors-24-01743-f002:**
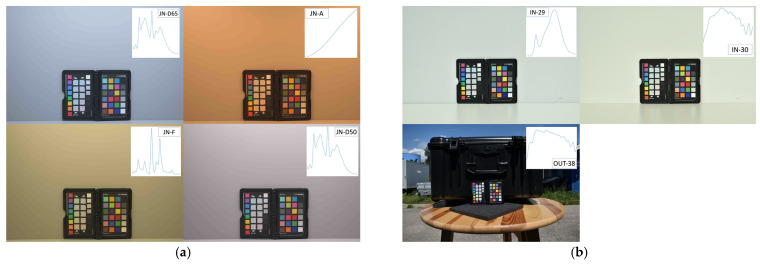
Nikon D5600 sample photos (displayed as fully rendered JPGs) of the XRCCPP under different light conditions (SPD of illuminant is showed at upper right for each image): (**a**) Laboratory settings: JN-D65, JN-A, JN-F, and JN-D50. (**b**) In situ settings: IN-29, IN-30, and OUT-38.

**Figure 3 sensors-24-01743-f003:**
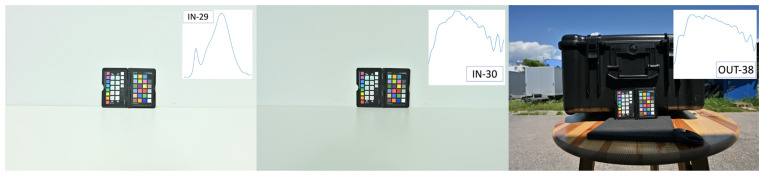
Nikon Z 7II sample photos (displayed as fully rendered JPGs) of the XRCCPP under in situ illuminants: IN-29, IN-30, and IN-38). SPD of illuminant is showed at upper right for each image.

**Figure 4 sensors-24-01743-f004:**
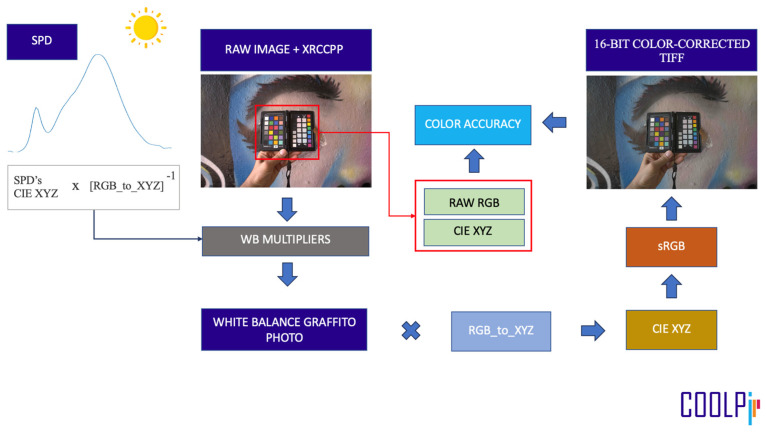
Color correction pipeline.

**Figure 5 sensors-24-01743-f005:**
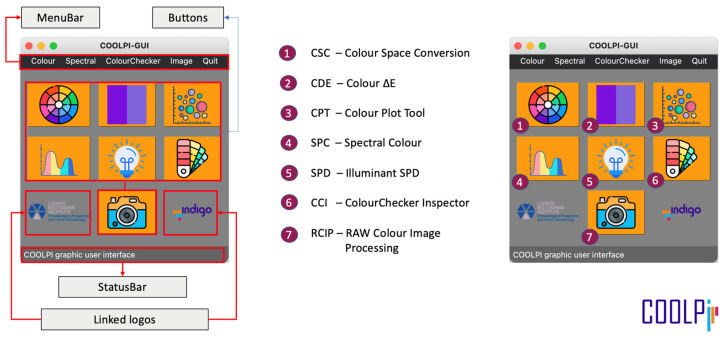
COOLPI-GUI initial view (Source: COOLPI API Documentation [28]).

**Figure 6 sensors-24-01743-f006:**
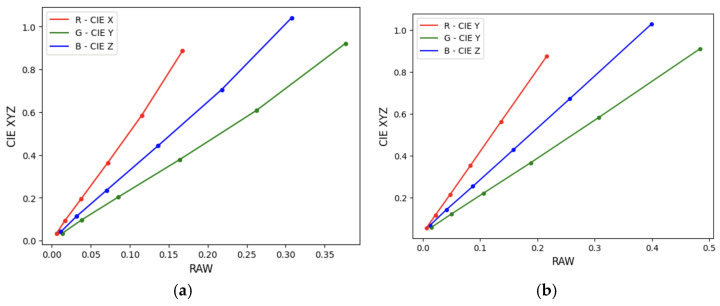
Assessment of the linearity of RAW data from the Nikon D5600 camera. Grayscale patch plot for two color reference targets: (**a**) XRCCPP. (**b**) CCDSG.

**Figure 7 sensors-24-01743-f007:**
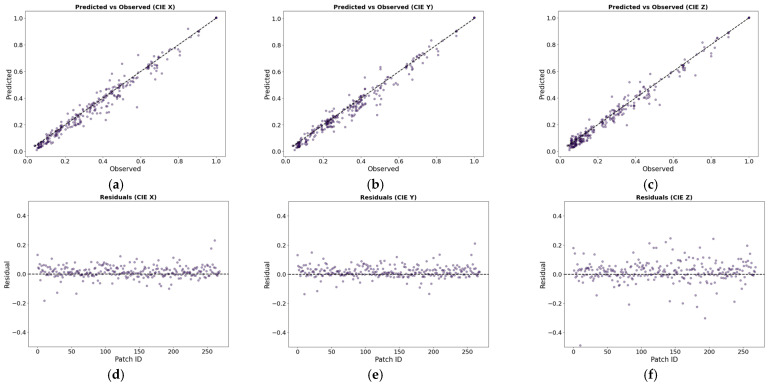
The upper row depicts the predicted versus observed CIE XYZ values for: (**a**) CIE X, (**b**) CIE Y, and (**c**) CIE Z. The lower row depicts the residuals: (**d**) CIE X, (**e**) CIE Y, (**f**) CIE Z. These values result from an OLS adjustment on all ND-WPPDS data.

**Figure 8 sensors-24-01743-f008:**
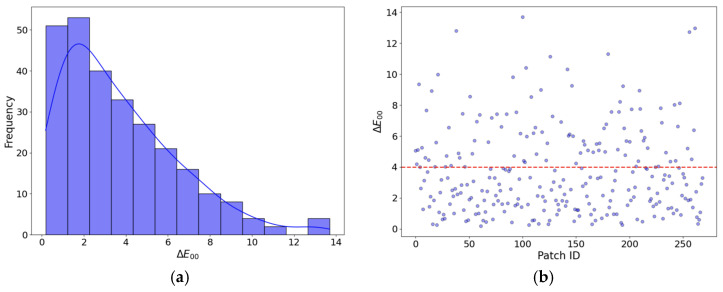
${∆E}_{00}$ color difference values after an OLS adjustment on all ND-WPPDS data: (**a**) Histogram; (**b**) values (the red horizontal line marks the ${∆E}_{00}$ limit set at four).

**Figure 9 sensors-24-01743-f009:**
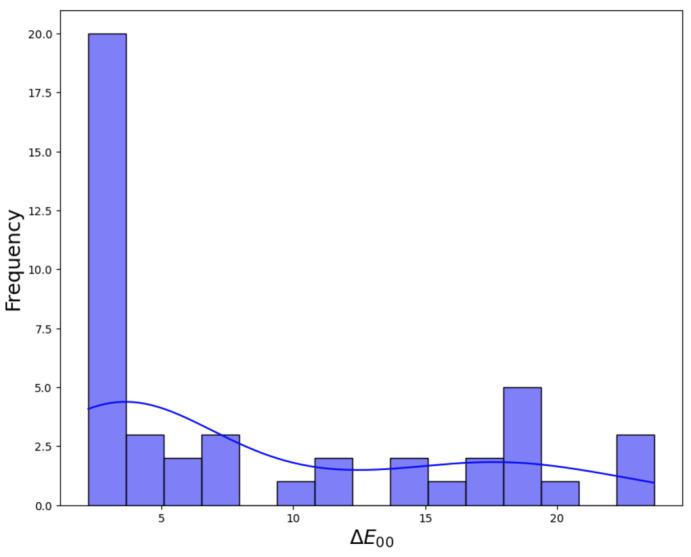
Histogram of ${∆E}_{00}$ color difference values for the graffiti sample images.

**Figure 10 sensors-24-01743-f010:**
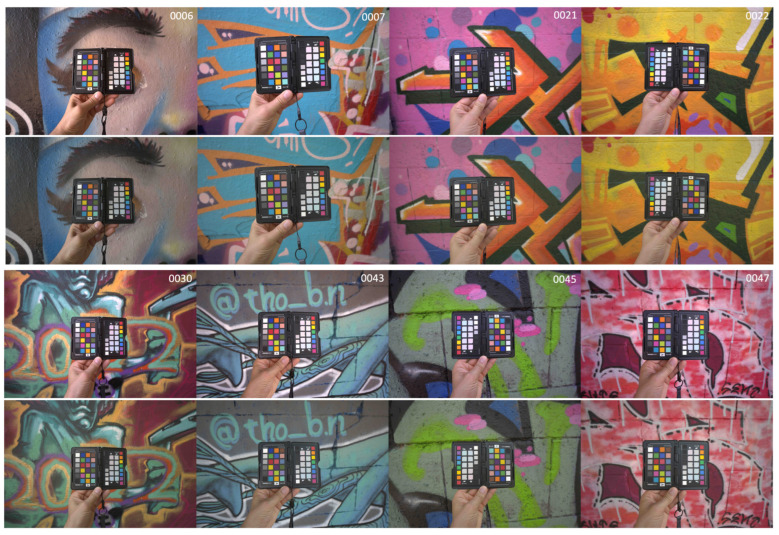
Application of the Nikon Z 7II transformation matrix to a sample of graffiti images. The upper photo of each pair is the in-camera-generated JPG photo, with its color-corrected version depicted below. The graffito ID is mentioned in the upper right corner of each photo pair.

**Figure 11 sensors-24-01743-f011:**
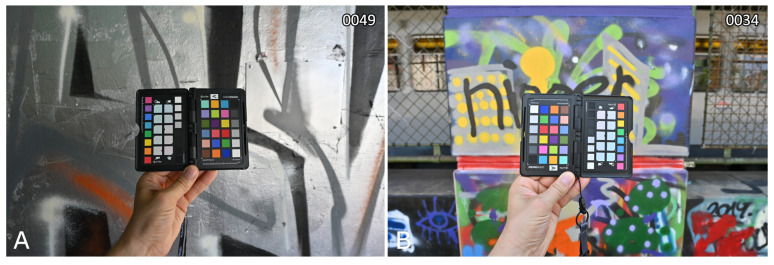
Two cases where the photographs of the XRCCPP target are not ideal. (Case **A**), for image of graffito take in a shadow area. (Case **B**), for graffito including metal and fluorescent inks. The graffito ID is mentioned in the upper right corner of each photo pair.

**Table 1 sensors-24-01743-t001:** Illuminant information: CCT in K and white point *X*_n_, *Y*_n_, and *Z*_n_ values (CIE 1931 2° standard observer).

Illuminant		CCT (K)	*X* _n_	*Y* _n_	*Z* _n_
Laboratory	JN-D65	6658	96.49	100	113.77
	JN-A	2681	112.11	100	30.53
	JN-F	3872	99.70	100	56.42
	JND50	4963	99.13	100	87.52
In situ	IN-29	3066	106.69	100	38.75
	IN-30	5027	93.60	100	76.96
	OUT-38	5557	96.41	100	94.72

**Table 2 sensors-24-01743-t002:** Datasets for model computation.

Camera	Dataset	Laboratory		In Situ		Total	
		Photos	Patches	Photos	Patches	Photos	Patches
Nikon D5600	ND-ODS	16	768	12	576	28	1344
	ND-WPPDS	16	768	12	576	28	1344
Nikon Z 7II	NZ-WPPDS	-	-	12	576	12	576

**Table 3 sensors-24-01743-t003:** Color reference target selection via an OLS model assessment on the ND-WPPDS dataset under JN-D65 illuminant (the best results are highlighted in bold).

Reference Target	Training Patches	Test Patches	RMSE	R^2^	Avg. Res. *X*	Avg. Res. *Y*	Avg. Res. *Z*	${∆E}_{ab}^{*}$	${∆E}_{00}$
CCDSG	76	20	0.046	0.952	0.020	0.023	0.020	6.571	4.143
CCC	19	5	0.027	0.766	−0.002	−0.004	0.009	4.037	3.198
XRCCPP	19	5	0.010	0.968	−0.003	−0.004	−0.005	2.254	**1.642**
SCK100	38	10	0.087	0.908	−0.020	−0.018	−0.011	6.678	4.540
All	153	39	0.067	0.909	0.012	0.012	0.021	6.562	4.808

**Table 4 sensors-24-01743-t004:** OLS model assessment metrics for the ND-ODS dataset only for the XRCCPP target under the different illuminants (highlighted in bold are the best results obtained).

Illuminant	Dataset	Training Patches	Test Patches	RMSE	R^2^	Avg. Res. *X*	Avg. Res. *Y*	Avg. Res. *Z*	${∆E}_{ab}^{*}$	${∆E}_{00}$
JN-D65	ND-ODS	19	5	0.009	0.968	−0.002	−0.003	−0.005	2.103	1.523
	NP-WPPDS	19	5	0.010	0.968	−0.003	−0.004	−0.005	2.254	1.642
JN-A	ND-ODS	19	5	0.008	0.967	−0.003	−0.004	−0.005	2.511	1.722
	NP-WPPDS	19	5	0.011	0.967	−0.006	−0.005	−0.021	3.518	1.854
JN-F	ND-ODS	19	5	0.008	0.965	−0.004	−0.004	−0.006	2.570	1.768
	NP-WPPDS	19	5	0.012	0.965	−0.006	−0.005	−0.014	2.917	2.000
JN-D50	ND-ODS	19	5	0.009	0.964	−0.004	−0.004	−0.006	2.419	1.730
	NP-WPPDS	19	5	0.011	0.964	−0.005	−0.005	−0.009	2.621	1.897
IN-29	ND-ODS	19	5	0.006	**0.971**	−0.001	0.000	−0.008	2.273	1.264
	NP-WPPDS	19	5	0.013	0.971	−0.003	0.000	−0.027	3.499	1.394
**IN-30**	ND-ODS	19	5	0.009	0.964	0.001	0.001	−0.008	2.189	**1.255**
	NP-WPPDS	19	5	0.014	0.964	0.001	0.001	−0.013	2.552	**1.272**
OUT-38	ND-ODS	19	5	0.010	0.967	0.002	0.002	−0.005	2.361	1.529
	NP-WPPDS	19	5	0.013	0.967	0.002	0.002	−0.006	2.594	1.584
Laboratory	ND-ODS	76	20	0.040	0.955	−0.005	−0.003	−0.001	6.746	4.201
	NP-WPPDS	76	20	0.037	0.976	−0.015	−0.014	−0.027	6.897	3.706
In situ	ND-ODS	57	15	0.042	0.904	−0.016	−0.009	−0.032	6.789	4.505
	NP-WPPDS	57	15	0.019	**0.989**	0.009	0.014	0.011	3.266	1.856
All	ND-ODS	134	34	0.048	0.927	−0.012	−0.012	−0.019	5.925	4.157
	NP-WPPDS	134	34	0.038	0.971	−0.005	−0.003	−0.026	5.896	3.279

**Table 5 sensors-24-01743-t005:** RGB-to-XYZ 3 × 3 transformation matrix coefficients for the XRCCPP color chart data.

Dataset	JN-D65			JN-A					
ND-ODS	3.5485	0.1360	0.4589	4.1801	0.4118	0.8701			
	1.4982	1.7705	−0.3188	1.5919	2.5406	−0.1602			
	0.5422	−0.8034	3.6980	0.1213	−0.5481	4.1887			
ND-WPPDS	0.6953	0.0594	0.1624	0.7623	0.0888	0.0709			
	0.2819	0.7426	−0.1084	0.3249	0.6133	−0.0146			
	0.0918	−0.3032	1.1309	0.0830	−0.4434	1.2797			
**Dataset**	**JN-F**			**JN-D50**					
ND-ODS	4.7730	0.1895	1.1060	4.1220	0.1895	0.5299			
	2.0052	2.3411	−0.0003	1.6664	2.1571	−0.3801			
	0.2961	−0.6924	5.0922	0.4070	−0.8313	4.2663			
ND-WPPDS	0.7297	0.0500	0.1404	0.7355	0.0641	0.1255			
	0.3046	0.6142	0.0000	0.2937	0.7212	−0.0889			
	0.0819	−0.3309	1.1703	0.0839	−0.3249	1.1664			
**Dataset**		**IN-29**			**IN-30**		**OUT-38**		
ND-ODS	2.8148	0.4314	0.1301	3.4889	0.4330	0.1308	4.5453	0.4733	0.3683
	1.0775	1.8053	−0.4953	1.3130	2.1511	−0.6684	1.7388	2.7641	−0.7463
	0.0871	−0.3682	2.8301	0.2026	−0.6176	3.6019	0.3684	−0.9187	4.9654
ND-WPPDS	0.8150	0.1728	0.0223	0.7815	0.2029	0.0391	0.7478	0.1628	0.0918
	0.3323	0.7703	−0.0904	0.2743	0.9397	−0.1861	0.2744	0.9119	−0.1783
	0.0708	−0.4137	1.3605	0.0564	−0.3593	1.3358	0.0632	−0.3293	1.2888
**Dataset**	**Laboratory**			**In situ**			**All**		
ND-ODS	4.4103	0.3033	0.0664	2.9184	0.4311	0.8140	3.6604	0.4111	0.3825
	1.9251	2.2850	−0.8959	0.9099	2.0879	−0.0174	1.4064	2.2066	−0.4543
	0.3476	−0.4924	3.62015	0.1157	−0.7326	4.1765	0.1273	−0.5518	3.9372
ND-WPPDS	0.7397	0.0774	0.1066	0.7877	0.1662	0.0511	0.7410	0.1079	0.1084
	0.3196	0.6702	−0.0671	0.3140	0.8335	−0.1426	0.2938	0.7331	−0.0683
	0.0874	−0.3511	1.1856	0.0737	−0.3717	1.3119	0.0612	−0.3385	1.2446

**Table 6 sensors-24-01743-t006:** OLS model assessment metrics for the full ND-WPPDS dataset (highlighted in bold are the best results obtained).

Illuminant	Training Patches	Test Patches	RMSE	R^2^	Avg. Res. *X*	Avg. Res. *Y*	Avg. Res. *Z*	${∆E}_{ab}^{*}$	${∆E}_{00}$
JN-D65	153	39	0.067	0.909	0.012	0.012	0.021	6.562	4.808
JN-A	153	39	0.056	0.943	0.009	0.006	0.040	8.342	4.796
JN-F	153	39	0.072	0.903	0.013	0.010	0.036	7.476	5.053
JN-D50	153	39	0.059	0.932	0.011	0.011	0.024	6.345	4.415
IN-29	153	39	0.054	0.946	0.007	0.008	0.022	4.980	2.844
IN-30	153	39	0.013	0.997	−0.001	0.000	−0.001	2.984	1.576
OUT-38	153	39	0.019	0.993	0.005	0.006	0.007	4.140	2.288
Laboratory	614	154	0.058	0.921	0.012	0.014	0.027	8.245	5.067
**In situ**	460	116	0.027	**0.985**	0.007	0.007	0.011	5.022	**2.810**
All	1075	269	0.043	0.966	0.015	0.017	0.017	6.071	3.698

**Table 7 sensors-24-01743-t007:** RGB-to-XYZ 3 × 3 transformation matrix coefficients for the full ND-WPPDS dataset.

	JN-D65			JN-A				
0.7136	0.3110	−0.0204	0.7712	0.3471	−0.1079			
0.2745	1.0495	−0.3168	0.3110	0.8950	−0.1935			
0.0781	−0.2069	1.1252	0.0825	−0.3772	1.2917			
	**JN-F**			**JN-D50**				
0.7470	0.2725	−0.0108	0.7463	0.2937	−0.0424			
0.2956	0.8833	−0.1671	0.2872	0.9863	−0.2718			
0.0733	−0.2568	1.1813	0.0824	−0.2562	1.1665			
	**IN-29**			**IN-30**			**OUT-38**	
0.7529	0.2819	−0.0277	0.7474	0.2329	0.0283	0.7454	0.2337	0.0398
0.3129	0.8172	−0.1206	0.2758	0.9174	−0.1790	0.2939	0.9459	−0.2171
0.1110	−0.4290	1.3255	0.0780	−0.3460	1.2788	0.0996	−0.3195	1.2443
**Laboratory**				**In situ**			**All**	
0.7616	0.2451	0.0008	0.7406	0.2569	0.0142	0.7524	0.2343	0.0180
0.3142	0.8767	−0.1829	0.2916	0.8828	−0.1597	0.3040	0.8636	−0.1607
0.1025	−0.3399	1.2343	0.0923	−0.3644	1.2854	0.0967	−0.3527	1.2582

**Table 8 sensors-24-01743-t008:** Model assessment metrics for the different linear regression adjustments for the full ND-WPPDS dataset (the best results obtained are highlighted in bold).

Model	Training Patches	Test Patches	RMSE	R^2^	Avg. Res. *X*	Avg. Res. *Y*	Avg. Res. *Z*	${∆E}_{ab}^{*}$	${∆E}_{00}$
OLS	1075	269	0.043	0.966	0.015	0.017	0.017	6.071	3.698
MTLCV	1075	269	0.043	0.966	0.016	0.017	0.018	6.216	3.771
BR	1075	269	0.043	0.966	0.015	0.017	0.017	6.069	3.698
**HR**	1075	269	0.043	**0.967**	0.013	0.015	0.014	5.991	**3.657**
TSR	1075	269	0.044	0.964	−0.002	0.001	−0.013	6.136	3.701

**Table 9 sensors-24-01743-t009:** RGB-to-XYZ 3x3 transformation matrix coefficients for the different linear regression adjustments for the full ND-WPPDS dataset.

	OLS			MTLCV				
0.7524	0.2343	0.0180	0.7449	0.2411	0.0188			
0.3040	0.8636	−0.1607	0.3187	0.8249	−0.1368			
0.0967	−0.3527	1.2582	0.0807	−0.3069	1.2266			
	**BR**			**HR**			**TSR**	
0.7522	0.2345	0.0180	0.7596	0.2223	0.0282	0.7920	0.2010	0.0596
0.3044	0.8625	−0.1600	0.3085	0.8510	−0.1475	0.3377	0.8449	−0.1272
0.0965	−0.3521	1.2577	0.0877	−0.3382	1.2608	0.1225	−0.3924	1.3269

**Table 10 sensors-24-01743-t010:** OLS model assessment metrics for the full NZ-WPPDS dataset (the best results are highlighted in bold).

Illuminant	Training Patches	Test Patches	RMSE	R^2^	Avg. Res. *X*	Avg. Res. *Y*	Avg. Res. *Z*	${∆E}_{ab}^{*}$	${∆E}_{00}$
IN-29	153	39	0.018	0.994	0.000	0.000	0.001	4.103	2.071
IN-30	153	39	0.016	0.995	−0.002	−0.002	−0.003	3.368	**1.797**
OUT-38	153	39	0.030	0.982	0.008	0.009	0.008	4.970	2.759
In situ	459	117	0.027	0.985	0.007	0.008	0.007	4.959	2.911

**Table 11 sensors-24-01743-t011:** RGB-to-XYZ 3 × 3 transformation matrix coefficients for the full NZ-WPPDS dataset.

IN-29			IN-30			OUT-38		
0.7580	0.2274	0.0119	0.7088	0.2467	0.0537	0.7243	0.2387	0.0646
0.3337	0.7386	−0.0726	0.2764	0.8522	−0.1159	0.3030	0.8829	−0.1553
0.0620	−0.3130	1.2507	0.0614	−0.2798	1.2335	0.0902	−0.2727	1.2178
**In situ**								
0.7237	0.2418	0.0452						
0.3015	0.8145	−0.1032						
0.0697	−0.2883	1.2330						

**Table 12 sensors-24-01743-t012:** ${∆E}_{00}$ color difference values for the graffiti sample images (instances with a difference of less than four are highlighted in bold).

Graffito ID	${∆E}_{00}$	Graffito ID	${∆E}_{00}$	Graffito ID	${∆E}_{00}$
0001	23.497	0017	17.853	0034	18.592
0002	15.315	0018	6.143	0035	12.128
0003	18.224	0019	18.241	0037	**3.700**
0004	16.729	0021	**3.074**	0038	**2.628**
0005	5.513	0022	**2.781**	0039	**3.297**
0006	**2.891**	0023	**2.332**	0040	**3.292**
0007	**3.630**	0024	4.548	0041	**2.980**
0008	**2.544**	0026	7.701	0042	7.594
0010	14.004	0027	7.787	0043	**2.307**
0011	14.921	0028	9.682	0044	**2.236**
0012	19.012	0029	**2.842**	0045	**3.444**
0013	**3.688**	0030	**2.563**	0046	**3.217**
0014	11.019	0031	**3.294**	0047	**3.192**
0015	23.226	0032	**3.654**	0048	23.671
0016	20.518	0033	**3.232**	0049	17.991

**Table 13 sensors-24-01743-t013:** All graffiti photos classified according to their ${∆E}_{00}$ color difference.

$\mathrm{Interval}\;{∆E}_{00}$	Graffiti Photos	Percentage	$\mathrm{Average}/\mathrm{Median}\;{∆E}_{00}$
(0, 4]	22	49	3.037/3.133
(4, 24)	23	51	14.518/15.315
Total	45	100	8.905/4.548

## Data Availability

The code for the analysis performed is available in the COOLPI GitHub repository at https://github.com/GraffitiProjectINDIGO/coolpi/tree/main/wpp_data (accessed on 24 February 2024).

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
