# Peer review of "Practical RGB-to-XYZ Color Transformation Matrix Estimation under Different Lighting Conditions for Graffiti Documentation"

_sensors, 2024, doi:10.3390/s24061743_

Round 1
Reviewer 1 Report
Comments and Suggestions for Authors
This manuscript investigates characterization techniques in heritage documentation.
1. It is technically good but some statistical analysis related to conversion of change of color domain. So, the authors should provide that analysis part.
2. Is the proposed method suitable for nonlinear models?
3. In line 156, what is 2^{0} observer?
4. Improve the quality of all figures.
5. Rearrange the lines 208-216 since there is a jump in the paragraph.
6. Need some clarification about tonal range in equation 1.
7. Any specific reason to fix kL = kC = kH = 1, provide if any.
Comments on the Quality of English Language
The author should polish the language.
Author Response
Authors appreciate the comments and suggestions and want to thank this referee for the efforts made reading and reviewing this paper.

Reviewer 2 Report
Comments and Suggestions for Authors
This manuscript describes a procedure to determine an RGB-to-XYZ color transformation matrix for graffiti documentation. The procedure that is described in itself is not entirely novel, it is comprised of various components from literature. The novelty lies in the analysis that is performed using a specific dataset in relation to the various options these components offer.
I think the most interesting part of the manuscript is the analysis if white point preservation in the computation of an RGB-to-XYZ transformation matrix results in accurate (or acceptable) results. In addition, this manuscript demonstrates the (expected) discrepancy between the results on data sets generated under controlled cirumstances and the results on real-world graffiti images. There are a few items to consider for a potential revision of this manuscript:
- Authors mention a threshold for color differences of 4 and refer to [11] for motivation. However, in [11] they refer to other articles to back up this threshold, amongst other authors ref [2], so please use the correct reference for the motivatio for certain thresholds / tolerances.
- The threshold of 4.0 is used in literature (e.g., [2]) as part of a standard for preservation imaging guidelines (Metamorfoze). However, this standard not only comprises an average of 4.0 but also a maximum color difference of 10.0. Hence, in authors' analysis, the maximum should be added in addition to the reported average color difference. M
- Although I appreciate authors showing both dEab and dE2000, I can't agree with the fact that tolerance-levels from dEab are applied to dE2000 (or vice versa). Color difference values dEab cannot directly be compared to color difference values dE2000, dEab-values can only be compared to other dEab-values and dE2000-values can only be compared to other dE2000-values. The tolerances that are being reported in [2] are on dEab, so please use dEab-values in relation to these tolerances.
- Moreover, I would appreciate if authors explicitly mention that their aim is to adhere to this standard because I find it hard to agree with their (repeated) statements that dE2000 color differences below 4.0 are imperceptible to experienced observers. Without context, this is simply not true. The context is given in the Metamorfoze standard, so the best way to proceed would be to mention that you are attempting to adhere to the "most demanding standard" [2, p.1] of Metamorfoze. Which uses dEab-tolerances which cannot directly be convert to dE2000 anyways.
- Regarding Section 3.2, Linear Regression Model Comparison, this part of the manuscript seems to me a bit "trial and error". Instead of arguing why certain other linear regression models might work better than OLS, authors seem to be just applying these models "because they are available in the Python toolbox". I'm not surprised by the results in Table 8 since typically such linear models will only result in significantly different results if the data set has certain peculiar properties that are better handled by some models than other. In my opinion, Section 3.2 adds little to the paper can could be summarized by a single sentence, e.g., "other linear regression models did not result in better results".
Author Response

(The authors gave the same response as above.)
